# Cardiac Fibrosis: Mechanistic Discoveries Linked to SGLT2 Inhibitors

**DOI:** 10.3390/ph18030313

**Published:** 2025-02-24

**Authors:** Filip Rolski, Michał Mączewski

**Affiliations:** Department of Clinical Physiology, Centre of Postgraduate Medical Education, 99/103 Marymoncka Str., 01-813 Warsaw, Poland; filip.rolski@cmkp.edu.pl

**Keywords:** heart failure, SGLT2 inhibitor, cardiac fibrosis

## Abstract

Sodium-glucose cotransporter 2 inhibitors (SGLT2is), commonly known as flozins, have garnered attention not only for their glucose-lowering effects in type 2 diabetes mellitus (T2DM) but also for their cardioprotective properties. This review examines the mechanisms underlying the anti-fibrotic effects of SGLT2is, with a focus on key clinical trials and preclinical models. SGLT2is, mainly empagliflozin and dapagliflozin, have demonstrated significant reductions in heart failure-related hospitalizations, cardiovascular death, and fibrosis markers, independent of their glucose-lowering effects. The cardioprotective benefits appear to stem from direct actions on cardiac tissues, modulation of inflammatory responses, and improvements in metabolic parameters. In animal models of heart failure, SGLT2is were demonstrated to reduce cardiac fibrosis through mechanisms involving AMPK activation, reduced oxidative stress, and inhibition of pro-fibrotic pathways, not only through the inhibition of SGLT2 present on cardiac cells but also by targeting several other molecular targets. These findings confirm their efficacy in the treatment of heart failure and align with evidence from human trials, supporting the potential involvement of multiple pathways in mediating cardiac fibrosis. These results also provide a promising basis for clinical trials specifically targeting pathways shared with SGLT2is.

## 1. Introduction

Sodium-glucose cotransporter 2 inhibitors (SGLT2is) are renowned for their primary role in managing hyperglycemia in type 2 diabetes mellitus (T2DM) [1]. More recently, the cardiovascular benefits of SGLT2is, particularly concerning cardiac fibrosis, are attracting considerable attention [2]. Investigating the mechanisms behind their diverse effects not only sheds light on their potential therapeutic applications beyond glycemic control but also opens new research pathways for treating adverse cardiac fibrosis and possibly fibrosis in other organs.

Cardiac fibrosis is a pathological condition characterized by the excessive deposition of extracellular matrix (ECM) components, such as collagen, in the heart tissue, leading to stiffening and impaired cardiac function [3]. It is a hallmark of various cardiovascular diseases (CVDs), including heart failure, myocardial infarction, and hypertensive heart disease. The progression of cardiac fibrosis is driven by complex interactions between cardiac cells, inflammatory mediators, and neurohormonal factors, alterations in gene expression mediated by microRNA culminating in a detrimental remodeling of the myocardium that can compromise cardiac performance and increase the risk of arrhythmias and heart failure [4,5]. Available treatment for cardiac fibrosis is limited and currently no anti-fibrotic drug has demonstrated clear efficacy in clinical studies [6].

Emerging evidence has highlighted the potential role of a novel class of glucose-lowering agents in modulating cardiac fibrosis [7,8,9,10]. SGLT2is are primarily used to manage hyperglycemia in patients with type 2 diabetes mellitus (T2DM) by promoting the renal excretion of glucose [1]. However, beyond their glucose-lowering effects, SGLT2is have demonstrated significant cardiovascular benefits, including reductions in the risk of hospitalization for heart failure and cardiovascular death, as observed in several major clinical trials [11,12,13,14,15,16].

The mechanisms underlying the cardioprotective effects of SGLT2is extend beyond glycemic control and are thought to involve direct actions on cardiac tissues, modulation of inflammatory responses, and improvements in metabolic parameters [17,18,19,20,21,22]. Specifically, SGLT2is have been shown to exert anti-fibrotic effects in preclinical models of cardiac fibrosis, suggesting their potential utility in mitigating the adverse structural remodeling associated with this condition [19,23,24,25,26]. Given the growing burden of heart failure and the limited effectiveness of current therapies in reversing established fibrosis, the exploration of SGLT2i molecular mechanisms of action in the context of cardiac fibrosis represents a promising avenue of research for targeted and more efficient management of fibrosis [27,28,29]. It is worth noting that SGLT2 expression has been confirmed in cardiomyocytes and cardiac endothelial cells, and its expression increases in the case of heart failure [30,31,32]. Interestingly, SGLT2 is also expressed in the liver, but its expression does not increase in the case of chronic liver disease [33], and it is not expressed in the lungs [34]. This is of importance, as preclinical studies have demonstrated that SGLT2is also reduce fibrosis in these organs in various pathologies [35]. These findings suggest that the anti-fibrotic mechanism of action of SGLT2is is most likely a result of their interaction with off-targets that are explored in Section 4 of this review. This new potential application of SGLT2is could prove beneficial not only for the treatment of cardiac fibrosis but also fibrosis of other organs. This review aims to provide an overview of the pharmacological actions of SGLT2is, and the emerging evidence supporting their potential benefits in cardiac fibrosis treatment, thereby setting the stage for further investigation into their potential therapeutic applications and further scientific investigations.

### Literature Search Strategy

Research was performed in multiple databases, including PubMed, Scopus, Web of Science and Google Scholar, to identify relevant studies published in the last 10 years. The search was conducted using a combination of keywords “SGLT2 inhibitor”, “cardiovascular”, “heart failure”, “fibrosis” and “fibroblast”. Additional sources were identified through citation tracking of key articles and reviews. Studies were included if they focused on the effects of SGLT2 inhibitors on the cardiovascular system or signaling pathways directly connected to the mechanism of action of SGLT2is and were published in peer-reviewed journals.

## 2. Animal Studies

In a rat model of myocardial infarction (MI), SGLT2 silencing using a lentiviral vector was associated with a reduction in cardiac fibrosis. Additionally, silencing SGLT2 in cardiac fibroblasts resulted in a mitigated response to TGF-β, decreasing their collagen production and proliferation [30]. Taken together, these data suggest that SGLT2 expression in the failing heart may be an important driver of pathological cardiac remodeling. Nevertheless, animals studies have shown that SGLT2is influence several important pathways in the heart [19,25,36,37,38,39], possibly influencing the process of cardiac fibrosis in a multifaceted manner. This section presents evidence from animal studies supporting the anti-fibrotic effects of several SGLT2 inhibitors.

Empagliflozin ameliorated interstitial fibrosis, which was accompanied by improved diastolic function, in a pig model of MI-induced HFrEF [7]. This reduced histological fibrosis was paralleled by smaller extracellular volume using T1 mapping in cMRI. The empagliflozin group showed lower levels of both biochemically determined collagen and reduced gene expression of type-1 collagen. Empagliflozin also demonstrated efficacy in a rat model of HFpEF, leading to improvement in cardiac function and reduction in cardiac fibrosis and stiffness [10]. Improvement in cardiac functions and reduction in cardiac fibrosis were also observed in a mouse model of Parkinson’s disease due to a reduction in mitophagy and improved mitochondrial integrity [40]. In diabetic KK-Ay mice, empagliflozin has been demonstrated to reduce myocardial fibrosis by inhibiting the TGF-β/Smad pathway and activating Nrf2/ARE signaling [22]. Reduction in hypertrophy through the inhibition of sodium–hydrogen antiporter 1 (NHE1) was also observed in a diabetic mouse model upon empagliflozin administration [41]. Empagliflozin treatment was demonstrated to be effective in mouse models of both HFrEF and HFpEF and improvement in cardiac functions could result in the absence of changes in circulating ketone bodies, cardiac ketone oxidation, or increased cardiac ATP production, but a reduction in inflammatory signaling was observed [36]. A reduction in cardiac fibrosis, improvement in cardiac function and reduction in pro-inflammatory markers was also demonstrated in a rat model of hypertensive heart failure [42].

Dapagliflozin reduced interstitial fibrosis in mice after MI as well as reducing the expression of collagen 2 (Col2), collagen 3 (Col3), matrix metalloproteinase 2 (Mmp2) and 9 (Mmp9) at the mRNA level; reduced myocardial fibrosis and the protein expression of collagens 1 and 3 in a rabbit model of HFrEF induced by aortic stenosis [43]; and achieved the same in a mouse model of arrhythmogenic cardiomyopathy [24] as well as in a diabetic rabbit model [23]. In a diabetic rabbit model, dapagliflozin improved left ventricular diastolic function and mitigated cardiac fibrosis. This effect was connected to the inhibition of serum and glucocorticoid-regulated kinase 1 (SGK1) signaling, reducing myocardial fibrosis, inflammation, and mitochondrial disruption. Dapagliflozin treatment in a mouse model of isoproterenol-induced HF led to an improvement in cardiac function that was associated with a reduction in perivascular fibrosis [44]. In BTBR mice with type 2 diabetes, treatment with dapagliflozin attenuated the elevated mRNA levels of collagen 1 and collagen 3 in the heart [21]. In a mouse model of MI, dapagliflozin treatment resulted in reduced fibrosis and improved cardiac function, possibly stemming from enhanced angiogenesis [45].

Canagliflozin therapy was associated with decreased interstitial and perivascular myocardial fibrosis in chronically ischemic tissue in swine [46] and in a rat model of HFpEF [47]. In another study conducted on a rat model of HFpEF, canagliflozin treatment significantly reduced interstitial cardiac fibrosis and inhibited pro-fibrotic Wnt pathway [48].

Luseogliflozin reduced cardiac hypertrophy and fibrosis in diabetic mice by inhibiting NHE-1 activity, leading to decreased transforming growth factor-β2 (TGF-β2) expression in cardiomyocytes. This effect was demonstrated in both high glucose-exposed mouse cardiomyocytes and db/db mice, suggesting a mechanism independent of direct SGLT2 expression in the heart [49].

The data from preclinical studies suggest that the reduction in myocardial fibrosis is a class effect (since it is exhibited by all tested SGLT2 inhibitors) and confirming it at the histological level, and predominantly involves the inhibition of pro-fibrotic pathways. The overview of effects directly or indirectly connected to cardiac remodeling that are associated with the use of SGLT2i are presented in Figure 1.

## 3. Clinical Trials

Four landmark, large randomized clinical trials have established that two SGLT2is, dapagliflozin and empagliflozin, reduce composite cardiovascular end-points in patients with both heart failure with reduced ejection fraction (HFrEF) [12,13] and heart failure with preserved ejection fraction (HFpEF) [11,50], which is a unique feature for heart failure therapies. However, while in HFrEF a pooled analysis of the EMPEROR-Reduced and DAPA-HF trials showed a significant reduction in all-cause mortality and cardiovascular mortality, the benefits in both HFpEF trials, EMPEROR-Preserved and DELIVER, was driven principally by a reduction in HF hospitalizations.

These results have prompted further investigations into the potential effects of SGLT2is on cardiac fibrosis as a possible mediator of these cardioprotective effects. The EMPA-TROPISM study revealed a reduction in myocardial fibrosis assessed using cardiac Magnetic Resonance Imaging (cMRI) in HFrEF patients treated with empagliflozin [7,8], although empagliflozin reduced fibrosis volume proportionally to the reduction in cardiomyocyte volume, so that the final percentage of fibrosis was unchanged. Moreover, empagliflozin reduced cMRI-assessed fibrosis in patients with diabetes mellitus and coronary artery disease [9]. A meta-analysis of five studies enrolling patients with HFrEF, diabetes and coronary artery disease confirmed that empagliflozin effectively reduces diffuse cardiac fibrosis, as measured by cMRI [51].

Furthermore, empagliflozin reduced blood markers of collagen turnover, procollagen type I carboxy-terminal propeptide (PICP) and a fragment of N-terminal type III collagen (PRO-C3) [13]. This effect was evident after 12 weeks of therapy, persisted for at least 1 year of therapy and its magnitude was identical in patients with HFrEF and HFpEF. Of note, empagliflozin did not affect markers of collagen degradation.

This human data concerning the effects of SGLT2i on fibrosis are mainly limited to empagliflozin and suggest that it indeed reduces myocardial fibrosis (assessed indirectly using cMRI) in various clinical conditions, and its concurrent reduction in blood markers of collagen synthesis indicates that reduced collagen production is a probable mediator of these effects.

## 4. Cellular Mechanisms of Anti-Fibrotic Effects Mediated by SGLT2i

This section covers findings from in vitro studies on the molecular mechanisms through which SGLT2is possibly exert their anti-fibrotic effects, which extend beyond glycemic control and may help better understand the outcomes observed in human trials. Primary findings on the molecular anti-fibrotic effects of SGLT2is, focusing on the key cell types involved in mediating cardiac fibrosis, are presented. A summary of the molecular targets and their effects are presented in Figure 2.

### 4.1. Anti-Fibrotic Effects in Fibroblasts

The anti-fibrotic effects of SGLT2 inhibitors on the heart and peri-coronary regions have been studied in various animal models. These beneficial effects seem to occur independently of changes in blood pressure or glucose levels. In vitro, cardiac fibroblasts from BTBR mice displayed higher mRNA levels of NLR Family Pyrin Domain-containing 3 (NALP3), apoptosis-associated speck-like protein containing a caspase recruitment domain (ASC), IL-1β, and caspase-1, indicating increased inflammation. Dapagliflozin significantly attenuated these mRNA levels in an 5′AMP-activated protein kinase-dependent (AMPK) manner, but independently of SGLT1, suggesting a specific anti-inflammatory pathway linked to AMPK activation [21]. The upregulation of AMPK activity is suggested to result from the inhibition of Complex I of the respiratory chain, leading to increases in cellular AMP or ADP [52]. Experiments on isolated rat cardiac fibroblasts have shown that dapagliflozin attenuated fibroblast activation and reduced collagen production in response to high glucose levels. This effect was mediated through the inhibition of SMAD4 upregulation via an AMPK-dependent pathway [19]. Empagliflozin has also demonstrated direct anti-fibrotic effects on human cardiac fibroblasts. It reduced the activation and collagen remodeling of human cardiac myofibroblasts in response to TGF-β1, promoting a quiescent phenotype [53]. In a rat MI model, empagliflozin treatment led to a significant reduction in scar formation and cardiac fibrosis which was accompanied by reduction in TGF-β1 and SMAD3 expression [54]. Inhibition of the pro-fibrotic properties of human atrial fibroblasts in response to empagliflozin was also shown to depend on NHE1 inhibition [25]. Dapagliflozin was shown to reduce the migration rate of cardiac fibroblasts isolated from HFrEF patients, while it did not affect the migration of fibroblasts from healthy hearts. Furthermore, treatment with dapagliflozin resulted in decreased production of IL-1β, IL-6, MMP-3 and MMP-6, and reduced phosphorylation of STAT3, whereas ERK1/2 and MEK1/2 remained unaffected [55]. The anti-fibrotic effects of dapagliflozin and empagliflozin on cardiac fibroblasts were demonstrated to be at least partially mediated by direct interactions with NHE1, the sodium–myoinositol cotransporter (SMIT) and the sodium–multivitamin cotransporter (SMVT) [25,56].

### 4.2. Anti-Fibrotic Effects Through Effect on Endothelial Cells

Endothelial-to-mesenchymal transition (EndMT) is a crucial cellular process implicated in the pathogenesis of cardiac fibrosis [57]. During EndMT, endothelial cells lose their characteristic markers and functions, such as the expression of vascular endothelial cadherin (VE-cadherin) and the ability to form tight junctions, and acquire mesenchymal traits, including increased motility and the expression of α-smooth muscle actin (α-SMA) and fibroblast-specific protein 1 (FSP1) [58]. This phenotypic transition contributes to the proliferation of fibroblasts and myofibroblasts within the cardiac tissue, leading to excessive deposition of ECM components, such as collagen. This remodeling of the cardiac extracellular matrix disrupts the structural integrity and function of the heart, ultimately contributing to heart failure [59].

The inhibition of EndMT by SGLT2is appears to be mediated by the same biological mechanism as observed in fibroblasts. Dapagliflozin was shown to inhibit EndMT mediated by glucose through AMPKα-dependent inhibition of SMAD3/TGF-β signaling. Moreover, dapagliflozin treatment also reduced collagen secretion by endothelial cells, which could also potentially contribute to a reduction in perivascular fibrosis [19]. Another study has demonstrated that the inhibitory effect of dapagliflozin on EndMT can be attributed to the activation of SIRT1 [44]. Treatment with empagliflozin was also shown to inhibit EndMT through the inhibition of TAK-1 and nuclear factor kappa-light-chain-enhancer of activated B cells (Nf-κB) phosphorylation [60]. Moreover, dapagliflozin was shown to stimulate angiogenesis and endothelial cell proliferation by direct binding to the pregnane X receptor (PXR) [45] and protect from endothelial dysfunction by inhibiting sarco(endo)plasmic reticulum calcium-ATPase 2 (SERCA2) [37].

### 4.3. Modulation of Inflammatory Responses

SGLT2is exhibit anti-inflammatory effects by reducing pro-inflammatory cytokine production and modulating leukocyte phenotypes in cardiac tissues. This reduction in inflammation may be crucial for the cardioprotective effects mediated by SGLT2i, as chronic inflammation is a key driver of heart failure and cardiac fibrosis [61].

In a mouse model of arrhythmogenic cardiomyopathy, dapagliflozin was shown to ameliorate cardiac fibrosis and inflammation by reverting the Hypoxia-Inducible Factor 2 alpha (HIF-2α) signaling pathway and inhibiting IκB kinase phosphorylation and NF-κB p65 transcriptional activity [24]. In a mouse model of experimental autoimmune myocarditis, empagliflozin was shown to protect cardiomyocytes from pyroptosis through NF-κB inhibition [38] and reduce inflammatory response by favoring M2 macrophage differentiation, which was associated with the inhibition of Signal Transducer and Activator of Transcription 3 (STAT3)T activation [62]. Another study has demonstrated that the treatment of macrophages with empagliflozin reduces TNF-α production and iNOS expression, promoting M2 differentiation in an AMPK-dependent manner [63]. Protection from cardiomyocyte death was observed both in vitro and in a mouse model of doxorubicin-induced cardiotoxicity with empagliflozin treatment. Improvements in cardiac function and a reduction in cardiac fibrosis were also noted, which were associated with the inhibition of nucleotide-binding oligomerization domain-like receptor family protein 3 (NLRP3)-, NF-κB- and MyD88-related pathways and the inhibition of pJNK/STAT3 signaling by empagliflozin [39,64]. In another study, dapagliflozin treatment was shown to elicit positive effects on doxorubucin-induced dilated cardiomyopathy by inhibiting NLRP3 inflammasome, decreasing p38-dependent toll-like receptor 4 (TLR4) expression [65]. In a rat model of myocardial infarction, dapagliflozin was also shown to attenuate cardiac fibrosis and promote the M2 macrophage phenotype, which was also demonstrated to be associated with a reduction in STAT3 phosphorylation [66]. Similar effects were observed in a model of viral myocarditis, and in this case dapagliflozin was also shown to reduce the expression of several pro-inflammatory cytokines [67]. On the other hand, in a mouse model of doxorubicin-mediated cardiac dysfunction, dapagliflozin was shown to exert protective effects by increasing STAT3 phosphorylation to control levels in cardiomyocytes [68]. Empagliflozin was also demonstrated to inhibit NLRP3 (nucleotide-binding domain-like receptor protein 3) inflammasome activation in a Ca^2+^-dependent manner in both isolated mouse hearts and human cardiomyocytes. These findings suggest that empagliflozin may exert cardioprotective functions by restoring optimal cytoplasmic Ca^2+^ levels in the heart [36]. Reductions in ICAM-1, VCAM-1, TNF-α, and IL-6 were also observed in the HFpEF myocardium of both humans and rats, which was associated with empagliflozin treatment. This effect depended on the improvement of the NO–sGC–cGMP cascade and PKGIα activity, achieved through reduced PKGIα oxidation and polymerization [69].

Another anti-fibrotic mechanism of action of SGLT2i may be associated with their effects on T lymphocytes. Both empagliflozin and dapagliflozin were shown to inhibit the differentiation of pro-inflammatory Th1 and Th17 cell populations while increasing the populations of anti-inflammatory T regulatory cells in mouse models of non-alcoholic fatty liver disease and diabetic kidney disease [70,71]. Accordingly, in vitro empagliflozin treatment of CD4^+^ T cells from patients with immune thrombocytopenia resulted in decreased Th1 and Th17 differentiation while favoring Treg differentiation. These observations were attributed to the inhibition of the mammalian target of rapamycin (mTOR) by empagliflozin [72]. Of note, interleukin 17 was shown to mediate cardiac fibrosis in various animal models [73,74,75]. Canagliflozin was also shown to inhibit mTOR activity in isolated human helper T cells, resulting in decreased T cell receptor signaling, T cell activation, and decreased production of pro-inflammatory cytokines [76]. Similar observations were made in T2DM patients, in whom empagliflozin treatment reduced circulating populations of Th17 and increased Tregs [77]. Therefore, it is likely that the promotion of an anti-inflammatory phenotype in T lymphocytes by SGLT2i may contribute to both the anti-fibrotic and beneficial cardiac effects observed in human trials.

### 4.4. Modulation of MicroRNA Profile

MicroRNAs (miRs) are endogenous, short, single-stranded RNAs approximately 20 nucleotides in length that negatively regulate gene expression at the post-transcriptional level. Recent data highlight their crucial role in the pathogenesis of atherosclerosis, cardiac hypertrophy and fibrosis. The pro-fibrotic pathways discussed in this review are known to be highly regulated by miRs [78,79]. In patients with HFpEF and diabetes, empagliflozin was shown to significantly reduce the expression of miR-21 and miR-91, both of which are known biomarkers of heart failure [80]. Notably, miR-91 inhibition was shown to be a promising target in preventing detrimental cardiac remodeling [81]. Inhibition of miR-21 due to empagliflozin treatment was also observed in a rat model of hyperglycemia and was associated with a reduction in TGF-β activation and cardiac fibrosis [82]. Ipragliflozin treatment in a rat model of heart failure has also revealed alterations in the profile of miR expression associated with cardiac hypertrophy [83]. Moreover, in rat cardiac fibroblasts, the expression of SGLT2 was shown to be directly regulated by miR-141 and to promote cardiac fibrosis [30]. In a mouse model of non-alcoholic fatty liver disease, empagliflozin was shown to downregulate miR-34a-5p in stellate cells and prevent liver fibrosis through the inhibition of TGF-β signaling. Interestingly, dapagliflozin treatment downregulated miR-125a-5p in a mouse model of depression and inhibited NLRP3 inflammasome activation [84]. Although these studies provide insights into the potential interplay between SGLT2 inhibitors and miRs in cardiovascular disease, the exact mechanisms remain under investigation. Further research is needed to elucidate the specific miRNAs involved and to determine how their modulation by SGLT2 inhibitors translates into clinical benefits for patients with cardiovascular conditions.

## 5. Indirect Cardioprotective Effects Mediated by SGLT2is

SGLT2 inhibitors exert several indirect cardioprotective effects that may possibly contribute to their potential in mitigating cardiac fibrosis. SGLT2is reduce cardiac workload, modulate metabolism and maintain electrolyte balance, while also minimizing oxidative stress and enhancing autophagy. This section covers cardioprotective effects that do not involve direct modulation of fibrotic pathways.

### 5.1. Promotion of Renal Glucose Excretion

SGLT2 inhibitors lower blood glucose by blocking glucose reabsorption in the proximal renal tubules, causing glycosuria. This action reduces plasma glucose and indirectly provides cardiovascular benefits by lessening hyperglycemia-related vascular damage and inflammation [85,86].

### 5.2. Reduction in Cardiac Workload and Blood Pressure

Through osmotic diuresis, SGLT2is promote glucose and sodium excretion, decreasing blood pressure and cardiac workload. This can alleviate heart failure symptoms and reduce myocardial stress, potentially slowing fibrosis progression [17,87].

### 5.3. Metabolic Modulation and Ketogenesis

SGLT2is shift metabolism from glucose to lipids, boosting ketone production. Ketones serve as an efficient fuel for the heart, especially during stress, and may reduce oxidative stress and fibrosis. Dapagliflozin also reduces epicardial fat, aiding cardiovascular risk reduction [88].

### 5.4. Electrolyte Balance and Renal Protection

Beyond glycosuria, SGLT2is increase sodium, potassium and magnesium excretion, easing fluid overload in heart failure and supporting renal function. Improved kidney health and electrolyte balance enhance cardiovascular stability [89,90].

### 5.5. Reduction in Oxidative Stress

SGLT2is lower reactive oxygen species (ROS) and bolster antioxidant defenses, protecting cardiac cells. Empagliflozin, for instance, decreases ROS by inhibiting Na^+^/H^+^ exchange. Enhanced mitochondrial function and AMPK activation contribute to reduced oxidative stress in diabetic and hypertensive patients [91].

## 6. Conclusions

SGLT2 inhibitors show high potential in the prevention and management of cardiac fibrosis in both animals and humans. Unveiling which aspects of the SGLT2i mechanism of action are responsible for these effects could lead to the development of a more selective and potent class of anti-fibrotic drugs. More research is required to confirm which, if any, beneficial effects of SGLT2is in heart failure treatment stem from their inhibition of SGLT2 in cardiac tissues and which are due to off-target actions. Full understanding of these aspects of SGLT2is may pave the way for more efficient and personalized therapies both in heart failure and other diseases.

## Figures and Tables

**Figure 1 pharmaceuticals-18-00313-f001:**
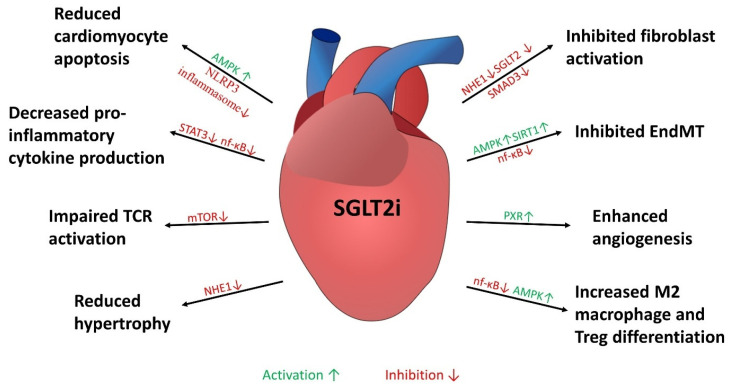
Cardioprotective effects mediated by SGLT2 inhibitors. Abbreviations; AMPK—5′AMP-activated protein kinase, EndMT—endothelial-to-mesenchymal transition, mTOR—mammalian target of rapamycin, NF-κB—nuclear factor kappa-light-chain-enhancer of activated B cells, NHE1—sodium/hydrogen exchanger 1, SIRT1—sirtuin 1, STAT3—signal transducer and activator of transcription 3, TCR—T cell receptor, Treg—T regulatory lymphocyte, PXR—pregnane X receptor.

**Figure 2 pharmaceuticals-18-00313-f002:**
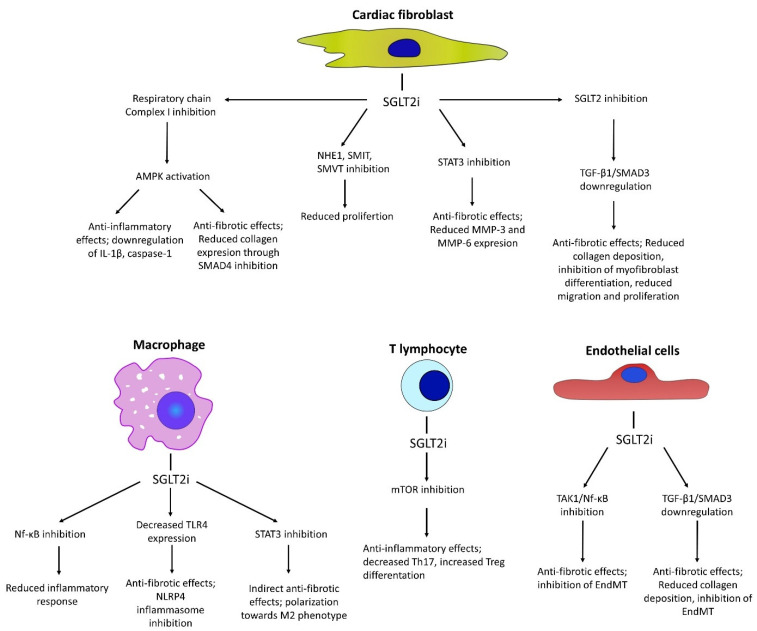
Molecular targets and effects mediated by SGLT2is in the context of cardiac fibrosis and inflammation. Abbreviations; AMPK—5′AMP-activated protein kinase, EndMT—endothelial-to-mesenchymal transition, mTOR—mammalian target of rapamycin, NF-κB—nuclear factor kappa-light-chain-enhancer of activated B cells, NHE1—sodium/hydrogen exchanger 1, SMIT—sodium/myo-inositol cotransporter, SMVT—sodium-dependent multivitamin transporter, STAT3—signal transducer and activator of transcription 3, MMP—matrix metalloproteinase, Treg—T regulatory lymphocyte.

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
