# Peer review of "Cardiac Fibrosis: Mechanistic Discoveries Linked to SGLT2 Inhibitors"

_pharmaceuticals, 2025, doi:10.3390/ph18030313_

Round 1

Reviewer 1 Report

Comments and Suggestions for Authors

This review paper, entitled: " Cardiac Fibrosis: Mechanistic Discoveries Linked to SGLT2 Inhibitors" is up-to-date, well-written, and comprehensive. The review addresses an important issue of the inhibition of cardiac fibrosis, a phenomenon very devastating heart, in particular left ventricle following cardiac damage.

The Authors have described most of mechanisms that underlie 

However, before the manuscript can be considered for publication, I believe two major issues should be addressed.

Major comments.

1) Introduction. The Authors have written that : "Cardiac fibrosis is a pathological condition characterized by the excessive deposition of extracellular matrix (ECM) components, such as collagen, in the heart tissue, leading to stiffening and impaired cardiac function". This is true, but it is too general.

I think that the Authors should explore this issue more, as cardiac fibrosis is driven by many factors, such as oxidative stress, inflammation, hypoxia, and genetic mechanisms involving microRNAs regulatory functions. In particular, microRNAs are involved in the excessive proliferation of fibroblasts, leading to  left ventricular remodeling and fibrosis. A great number of anti-, and pro-proliferative microRNAs are involved in this process. The Authors might find utile to cite the following paper: https://doi.org/10.3390/jcm11226849

Some of them are associated with SGLT2 pathways. For instance, miR-30 family may be involved in mediating SGLT2is' effect at the epigenetic level in endothelial cells in a synchronous manner. This interplay may result in preventing adverse cardiac events and pathological LV remodeling (doi: 10.1016/j.gendis.2023.101174). microRNAs act through all pathway mentioned in Figure 3, having impact on inflammatory mechanisms. In genetic and pharmacological in vitro and in vivo models, miR-132 (acting through NLRP3 inflammasome activation) has been shown to induce cardiac fibrosis. Recent studies have reported the inhibitory effects of anti-miR-132 on fibroblasts proliferation in cardiac animal models, and the safety and potential first efficacy signals of anti-miR-132 in patients with heart failure.

Therefore I believe that at least a short paragraph on microRNAs should be added to this interesting and important review paper.

2) SGLT2 inhibitor dapagliflozin reduces endothelial dysfunction and microvascular damage during cardiac ischemia/reperfusion injury through normalizing the XO-SERCA2-CaMKII-coffilin pathways. However, I feel that the mechanisms of action of SGLT2i should be explored more carefully with regard to perfusion in small vessels and capillaries, which is particularly important in patients with lower extremity arterial disease (PAOD).

Author Response

This review paper, entitled: " Cardiac Fibrosis: Mechanistic Discoveries Linked to SGLT2 Inhibitors" is up-to-date, well-written, and comprehensive. The review addresses an important issue of the inhibition of cardiac fibrosis, a phenomenon very devastating heart, in particular left ventricle following cardiac damage.

The Authors have described most of mechanisms that underlie 

Thank you for this favorable opinion.

However, before the manuscript can be considered for publication, I believe two major issues should be addressed.

Major comments.

1) Introduction. The Authors have written that : "Cardiac fibrosis is a pathological condition characterized by the excessive deposition of extracellular matrix (ECM) components, such as collagen, in the heart tissue, leading to stiffening and impaired cardiac function". This is true, but it is too general.

I think that the Authors should explore this issue more, as cardiac fibrosis is driven by many factors, such as oxidative stress, inflammation, hypoxia, and genetic mechanisms involving microRNAs regulatory functions. In particular, microRNAs are involved in the excessive proliferation of fibroblasts, leading to  left ventricular remodeling and fibrosis. A great number of anti-, and pro-proliferative microRNAs are involved in this process. The Authors might find utile to cite the following paper: https://doi.org/10.3390/jcm11226849

Some of them are associated with SGLT2 pathways. For instance, miR-30 family may be involved in mediating SGLT2is' effect at the epigenetic level in endothelial cells in a synchronous manner. This interplay may result in preventing adverse cardiac events and pathological LV remodeling (doi: 10.1016/j.gendis.2023.101174). microRNAs act through all pathway mentioned in Figure 3, having impact on inflammatory mechanisms. In genetic and pharmacological in vitro and in vivo models, miR-132 (acting through NLRP3 inflammasome activation) has been shown to induce cardiac fibrosis. Recent studies have reported the inhibitory effects of anti-miR-132 on fibroblasts proliferation in cardiac animal models, and the safety and potential first efficacy signals of anti-miR-132 in patients with heart failure.

Therefore I believe that at least a short paragraph on microRNAs should be added to this interesting and important review paper.

Thank you for your valuable suggestion. We have added a paragraph regarding this topic at the end of section 4 (highlighted in red). Although data is limited, we were able to find both clinical and pre-clinical studies showing that indeed SGLT2 inhibitors influence micro RNA involved in mediating cardiovascular disease.

2) SGLT2 inhibitor dapagliflozin reduces endothelial dysfunction and microvascular damage during cardiac ischemia/reperfusion injury through normalizing the XO-SERCA2-CaMKII-coffilin pathways. However, I feel that the mechanisms of action of SGLT2i should be explored more carefully with regard to perfusion in small vessels and capillaries, which is particularly important in patients with lower extremity arterial disease (PAOD).

Thank you for this very important comment. This is a very interesting topic, in fact we are currently working on the effects of SGLT2i on coronary endothelium. However, we believe that this goes beyond the area covered in this review paper, i.e. fibrosis, and we would like not to expand the review too much.

Reviewer 2 Report

Comments and Suggestions for Authors

I reviewed a manuscript titled “Cardiac Fibrosis: Mechanistic Discoveries Linked to SGLT2 Inhibitors”

This is a mini-review that seeks to highlight the mechanisms underlying the antifibrotic

effects of SGLT2i, with a focus on key clinical trials and preclinical models.

Consider the following when revising your manuscript. 

SGLT2i in the beginning of the introduction must be removed

1. the transition word "however" seems to be misplaced. Perhaps you can use "Additionally" or rephrase the second statement, and this must be supported by references

2. As this statement, "Emerging evidence has highlighted the potential role of a novel class of glucose-lowering agents in modulating cardiac fibrosis", there should be references to support this.

3. The over-emphasis/use of "in the context of" is distorting the flow of information in this manuscript. Remove or rather rephrase.

4. Throughout the manuscript and figures, revise the use of beta cell (B) as the symbol (β)

5. Although this is not a systematic review, it would offer some more insight to provide a section on how authors gathered evidence synthesized in this narrative review.

6. "It is a hallmark of various cardiovascular diseases (CVDs), including heart failure, myocardial infarction, and hypertensive heart disease." The statement is not referenced as this is not the author's new idea

7. This statement lack references "Nevertheless, animals studies have shown, that SGLT2i influence several important pathways in the heart, possibly influencing the process of cardiac fibrosis in a multifaceted manner."

8. Replace chapter with section

9. What is cardiomyopathy (19) ?

10. "In a diabetic rabbit model, dapagliflozin improved left ventricular diastolic function and mitigated cardiac fibrosis." no reference

11. In a mouse model MI, include "of"

12. "Thus animal studies support and extend data obtained in humans, suggesting that reduction of myocardial fibrosis is a class effect (since it is exhibited by all tested SGLT2 inhibitors) and confirming it at the histological level, and involves predominantly inhibition of pro-fibrotic pathways." This statement, especially the first statement made, is well written; however scientific flow is not clear.  Experiment researchers would start with invitro, invivo, preliminary trials, and then full clinical trials to confirm the results of the preclinical studies (invivo and invitro) and not a visa. Therefore, I would advise that authors re-structure their review, starting with animal evidence followed by clinical evidence. Then, you can compare both and explain if there is translatability of preclinical evidence in human/ clinical studies.

13. The conclusion is written as a discussion with references. I suggest you revise your conclusion to make a brief summary of what you have found in this review. Make your own conclusion without inserting references.

Author Response

This is a mini-review that seeks to highlight the mechanisms underlying the antifibrotic effects of SGLT2i, with a focus on key clinical trials and preclinical models.

 Consider the following when revising your manuscript. 

SGLT2i in the beginning of the introduction must be removed

  1. the transition word "however" seems to be misplaced. Perhaps you can use "Additionally" or rephrase the second statement, and this must be supported by references

 Thank you for your comment. This has been corrected.

  1. As this statement, "Emerging evidence has highlighted the potential role of a novel class of glucose-lowering agents in modulating cardiac fibrosis", there should be references to support this.

Thank you for your comment. References were added.

  1. The over-emphasis/use of "in the context of" is distorting the flow of information in this manuscript. Remove or rather rephrase.

Thank you for your valuable suggestion. These sections were rephrased

  1. Throughout the manuscript and figures, revise the use of beta cell (B) as the symbol (β)

Thank you for this comment. We checked carefully, but beta cell in each case refers to lymphocytes B, not pancreatic beta cells, therefore we left the “B” symbol to avoid confusion.

  1. Although this is not a systematic review, it would offer some more insight to provide a section on how authors gathered evidence synthesized in this narrative review.

Thank you for your suggestion. A paragraph regarding this topic has been added at the end of the introduction (highlighted in red).

  1. "It is a hallmark of various cardiovascular diseases (CVDs), including heart failure, myocardial infarction, and hypertensive heart disease." The statement is not referenced as this is not the author's new idea

Thank you for pointing this out. We have now added appropriate references to support this statement.

  1. This statement lack references "Nevertheless, animals studies have shown, that SGLT2i influence several important pathways in the heart, possibly influencing the process of cardiac fibrosis in a multifaceted manner."

Thank you for your observation. References were added.

  1. Replace chapter with section

Thank you for your suggestion. We have replaced 'chapter' with 'section' as recommended. 

  1. What is cardiomyopathy (19) ?

Thank you for pointing this out. This was a wrongly formatted citation and has now been corrected.

  1. "In a diabetic rabbit model, dapagliflozin improved left ventricular diastolic function and mitigated cardiac fibrosis." no reference

Thank you for your observation. A reference was added.

  1. In a mouse model MI, include "of"

Thank you for pointing this out. This was corrected.

  1. "Thus animal studies support and extend data obtained in humans, suggesting that reduction of myocardial fibrosis is a class effect (since it is exhibited by all tested SGLT2 inhibitors) and confirming it at the histological level, and involves predominantly inhibition of pro-fibrotic pathways." This statement, especially the first statement made, is well written; however scientific flow is not clear.  Experiment researchers would start with invitro, invivo, preliminary trials, and then full clinical trials to confirm the results of the preclinical studies (invivo and invitro) and not a visa. Therefore, I would advise that authors re-structure their review, starting with animal evidence followed by clinical evidence. Then, you can compare both and explain if there is translatability of preclinical evidence in human/ clinical studies.

Thank you for your thoughtful comment. We appreciate the standard scientific progression from in vitro and in vivo studies to clinical trials. However, in this particular case, the initial reports of benefits in heart failure treatment were observed in clinical trials before being explored in animal models. The preclinical studies were subsequently conducted to investigate the underlying mechanisms and confirm the histological changes. Given this sequence of discovery, we believe that our current structure, which presents clinical findings first and then supports them with preclinical data, more accurately reflects the progression of research in this field. We hope this explanation clarifies our approach, but we are open to further discussion if needed.

  1. The conclusion is written as a discussion with references. I suggest you revise your conclusion to make a brief summary of what you have found in this review. Make your own conclusion without inserting references.

Thank you for this comment. We have revised the conclusion section. The more relevant part has been moved to the introduction on page 2, and we have decided to delete the section regarding the effects of other classes of cardiovascular drugs on fibrosis. Please find the changes highlighted in red.

Round 2

Reviewer 1 Report

Comments and Suggestions for Authors

Dear Authors, I would like to congratulate you on this comprehensive and interesting review paper. 

Best regards 

Author Response

Thank you for your kind evaluation.

Reviewer 2 Report

Comments and Suggestions for Authors

The manuscript has been revised.

Ensure the text about animal evidence Somes before clinical evidence in text 

Author Response

Ensure the text about animal evidence comes before clinical evidence in text. 

Thank you for pointing out this issue. The Section 2 "Animal studies" now comes before the Section 3 "Clinical trials"